# TGF-β2 Induces Epithelial–Mesenchymal Transitions in 2D Planer and 3D Spheroids of the Human Corneal Stroma Fibroblasts in Different Manners

**DOI:** 10.3390/biomedicines11092513

**Published:** 2023-09-12

**Authors:** Araya Umetsu, Yosuke Ida, Tatsuya Sato, Masato Furuhashi, Hiroshi Ohguro, Megumi Watanabe

**Affiliations:** 1Department of Ophthalmology, Sapporo Medical University, Sapporo 060-8556, Japan; araya.umetsu@sapmed.ac.jp (A.U.); ooguro@sapmed.ac.jp (H.O.); 2Department of Cardiovascular, Renal and Metabolic Medicine, Sapporo Medical University, Sapporo 060-8556, Japan; 3Department of Cellular Physiology and Signal Transduction, Sapporo Medical University, Sapporo 060-8556, Japan

**Keywords:** three-dimensional spheroid culture, TGF-β, human corneal stroma fibroblasts (HCSFs), corneal injury model

## Abstract

To examine the epithelial–mesenchymal transition (EMT) that is induced on the human corneal stroma, two- and three-dimensional (2D and 3D) cultures of human corneal stroma fibroblasts (HCSFs) were used. In this study, HCSF 2D monolayers and 3D spheroids were characterized by (1) scanning electron microscopy (SEM), (2) trans-endothelial electrical resistance (TEER) measurements and fluorescein isothiocyanate (FITC)-dextran permeability, (3) cellular metabolic measurements, (4) the physical properties of 3D HCSF spheroids, and (5) the extracellular matrix (ECM) molecule gene expressions, including collagen (*COL*) *1*, *4 and 6*, and fibronectin (*FN*), a tissue inhibitor of metalloproteinase (*TIMP*) *1–4*, matrix metalloproteinase (*MMP*) *2*, *3*, *9 and 14*, and several endoplasmic reticulum (ER) stress-related factors. In the 2D HCSFs, TGF-β2 concentration-dependently generated (1) a considerable increase in ECM deposits revealed by SEM, (2) an increase in TEER values and a decrease in FITC-dextran permeability, (3) increases in both mitochondrial and glycolytic functions, and a substantial upregulation of *COL1*, *COL4*, *FN*, *αSMA*, *TIMP1*, TIMP, and most ER stress-related genes and the downregulation of *COL6* and *MMP3*. In the case of 3D spheroids, TGF-β2 induced the downsizing and stiffening of 3D spheroids and the upregulation of *COL6*, *MMP14*, and most ER stress-related genes. These findings suggest that TGF-β2 significantly induced a number of EMT-associated biological events including planar proliferation, cellular metabolic functions, and the production of ECM molecules in the 2D cultured HCSF cells, but these effects were significantly less pronounced in the case of 3D HCSF spheroids.

## 1. Introduction

In the space of injured corneal stroma, it is well known that the keratocytes change to spindle-shaped fibroblasts and acquire a migratory phenotype with increased actin expression. This increment in actin expression enables fibroblasts to proliferate and migrate toward the wound, repopulating the region that had been depleted of keratocytes through apoptosis [1,2,3]. Upon arrival to the injured corneal stroma, the fibroblasts differentiate into myofibroblasts that elaborate the extracellular matrix (ECM) and generate contractile forces engaged in corneal wound closure [4]. Clinically, such corneal stromal recovery from corneal damages was indeed observed in some cases surgically treated using simple limbal epithelial transplantation [5]. In addition, within biobanked eyes, the molecular or biomechanical properties of human donor corneal stroma can be restored by rehydration after a dehydrated condition for up to 2 weeks [6].

Recently, the epithelial–mesenchymal transition (EMT) and its reverse process, the mesenchymal–epithelial transition (MET), have been shown to have critical roles in these corneal wound healing [7] and dystrophy [8]. Transforming growth factor-β (TGF-β) signaling is one of the responsible pathways that induce EMT [9,10,11,12], thereby promoting corneal scarring processes. In fact, TGF-β affects the corneal stromal fibroblasts and transforms them into the corneal myofibroblasts that are characterized by the increase of α-smooth muscle actin (αSMA) [13]. It is also well known that αSMA is a marker of myofibroblast phenotype, especially as it relates to biological cues [14]. After proper corneal healing, myofibroblasts disappear from the wound space through apoptosis [15]. However, several reports document long-term corneal opacity from excessive numbers and/or prolonged persistence of myofibroblasts after healing [16,17]. Since myofibroblasts are less transparent than keratocytes and produce a disorganized ECM, this corneal stromal cell transformation induced corneal stromal opacity and fibrosis, but, in turn, could be a possible therapeutic strategy to prevent corneal stromal fibrosis.

It is shown that in addition to three TGF-β isoforms (TGF-β-1, -2, and -3), the TGF-β family members include activin, nodal, bone morphogenetic proteins (BMPs), growth and differentiation factors (GDFs), and other factors [18]. Among these TGF-β family members, it has been reported that the three isoforms of TGF-β, TGF-β-1, TGF-β-2, and TGF-β-3, are detected within ocular tissues including the cornea [19,20], conjunctival fibroblasts [21], and others [22]. Furthermore, their TGF-β receptors, that is, TGF-β R-1, R-2, and R-3, are also detected within human corneal endothelial cells [23] and conjunctival fibroblasts [21], suggesting that these three TGF-β isoforms are indeed involved in the pathophysiological roles within these ocular tissues. Among these three TGF-β isoforms, it has been shown that the roles of TGF-β1 and TGF-β3 have been vigorously characterized within the regulation mechanisms of corneal fibrosis [24,25,26]. In contrast, the pathophysiological contributions of TGF-β2 have been insufficiently investigated. Recently, it has also been shown that these three TGF-β isoforms are all detected within aqueous humor (AH) [27], suggesting their contribution to the pathogenesis of glaucoma [28,29,30], and among these, AH levels of TGF-β2 were most abundant in patients with glaucoma as well as healthy subjects [28]. Since for survival of the transparent corneal tissues, nutrients, oxygen, and various factors are exclusively required by perfusion from tear and AH, in which levels of TGF-β2 are significantly high, it is of great interest to study the TGF-β2 induced effects toward corneal stroma, especially their EMT.

Therefore, in the current study, to characterize the TGF-β2-induced corneal stromal EMT, our recently established in vitro two-dimensional (2D) and three-dimensional (3D) spheroid culture models using human corneal stromal fibroblasts (HCSF) [31,32] were subjected to following analyses: (1) ultrastructure by scanning electron microscopy (SEM) (2D); (2) trans-endothelial electrical resistance (TEER) measurements and fluorescein isothiocyanate (FITC)-dextran permeability (2D); (3) real-time cellular metabolic measurement (2D); (4) the physical properties of 3D spheroids, i.e., size and stiffness, respectively, and the gene expressions of extracellular matrix (ECM) molecules, their modulator, and endoplasmic reticulum (ER) stress-related factors (2D and 3D). 

## 2. Materials and Methods

The current study was authorized by the institutional review board (IRB registration number 282-8) in accordance with the principles of the Declaration of Helsinki and national regulations for the protection of personal data, and it was carried out at the Sapporo Medical University Hospital in Japan. All subjects who took part in this study provided their informed consent.

### 2.1. Preparation of 2D and 3D Spheroid Cultures of Human Corneal Stroma Fibroblasts (HCSFs)

HCSFs were isolated from surgically dissected human corneal stroma samples obtained from 2 patients with traumatic perforating injuries according to a previously described method [13]. The HCSF cells were then cultured and maintained in 150 mm 2D planar culture dishes until reaching 90% confluence at 37 °C in a 2D growth medium composed of HG-DMEM containing 10% FBS, 1% L-glutamine, and 1% antibiotic-antimycotic by daily medium exchange. The 3D spheroid culture of the HCSFs was processed using a hanging-droplet spheroid three-dimensional (3D) culture system as reported previously [31,32,33]. In brief, 2D cultured HCSFs were washed in phosphate-buffered saline (PBS) and then detached with 0.25% trypsin/EDTA. The cell pellet was re-suspended in a 3D spheroid medium made up of a 2D growth medium supplemented with 0.25% methylcellulose (Methocel A4M) after centrifugation at 300× *g* for 5 min. On day 0, each well of the hanging drop culture plate (#HDP1385, Sigma-Aldrich, St. Louis, MO, USA) received approximately 20,000 HCSFs in a 28 μL volume of 3D spheroid medium. On day 1, TGF-β2 (302-B2-002, R&D) was given to the 3D spheroids and continued until day 6 in various quantities (0, 1, or 5 nM). On each succeeding day, a new 14 μL of the medium was added in its place. These 3D HCSF spheroids, which had been cultivated under various circumstances as mentioned above, were harvested on day 6 and subjected to a number of studies, which are detailed below.

### 2.2. Trans-Endothelial Electron Resistance (TEER) Measurement, Ultrastructure by Scanning Electron Microscope (SEM), and Fluorescein Isothiocyanate (FITC)-Dextran Permeability of the 2D Cultured HCSF Monolayer

The 2D cultured HSCF cell monolayers were measured by TEER using the procedures previously reported [32]. Briefly, 2.0 × 10^4^ cells of 2D cultured HCSFs per well were seeded on 12 well plates for TEER (0.4 μm pore size and 12 mm diameter; Corning Transwell, Sigma-Aldrich, St. Louis, MO, USA). The apical and basal sides of each well were kept in 0.5 mL and 1.5 mL of growth media, respectively. The apical side was kept inside the membrane inserts, and the basal side was kept outside the membrane inserts. TGF-β2 was injected into the apical chamber at various dosages (0, 1, or 5 nM) when the cells were about 80% confluent (day 1). The culture medium was changed every other day. When the cells had reached 100% confluence on day 6, after two PBS washes, TEER (Ωcm^2^) values were determined using an electrical resistance system (KANTO CHEMICAL CO. INC., Tokyo, Japan) in accordance with the manual. Moreover, we performed scanning electron microscopy (EM) on HCSF cells on the membrane by using a HITACHI S-4300 microscope operated at 5 keV (the detector features 1280 × 960 pixels).

In order to test the fluorescein isothiocyanate (FITC)-dextran permeability, the basal compartments of the culture wells were coated with a 50 mol/L solution of FITC-dextran (Sigma-Aldrich, St. Louis, MO, USA), and after 60 min, the culture media from the upper compartment was gathered. Using a multifunctional plate reader (Enspire; Perkin Elmer, Waltham, MA, USA) with an excitation wavelength of 490 nm and an emission wavelength of 530 nm, the concentrations of the FITC-dextran were determined. The background concentration was calculated by using the fluorescence intensity of the control medium.

### 2.3. Measurement of Real-Time Cellular Metabolic Functions

As previously mentioned, the Seahorse XFe96 Bioanalyzer (Agilent Technologies, Santa Clara, CA, USA) was employed to measure the oxygen consumption rate (OCR) and extracellular acidification rate (ECAR) of 2D HCSFs in accordance with the manufacturer’s instructions with a few minor adjustments [34,35]. Briefly, 20,000 2D HCSFs that were untreated with or treated with TGF-β2 as described above were placed in wells of an ordinary Seahorse 96-well assay plate (#103794-100, Agilent Technologies, Santa Clara, CA, USA) before the day of assay, and the assay plate was incubated at 37 degrees Celsius. The culture medium was replaced with Seahorse XF DMEM assay medium (pH 7.4, #103575-100, Agilent Technologies) on the day of the test, and the assay plate was incubated for an hour at 37 °C in a CO_2_-free incubator with 5.5 mM glucose, 1.0 mM sodium pyruvate, and 2.0 mM glutamine. OCR and ECAR measurements were performed by using a Seahorse XFe96 Bioanalyzer at the baseline, and the injections of 2.0 M oligomycin, 5.0 M carbonyl cyanide p-trifluoromethoxyphenylhydrazone (FCCP), a mixture of 1.0 M rotenone and 1.0 M antimycin A, and 10 mM 2-deoxyglucose (2-DG) were made sequentially into the samples. The OCR and ECAR values were standardized by adjusting the amounts of protein assessed by a BCA protein assay (TaKaRa) per well by lysing the cells of the wells in which the measurements were completed with 10 μL of CelLytic^TM^ MT Cell Lysis Reagent (Sigma-Aldrich, St. Louis, MO, USA).

### 2.4. Quantitative PCR

Using an RNeasy mini kit (Qiagen, Valencia, CA, USA), total RNA was extracted from 2D-grown cells within a single well out of 12 wells of the culture dish or 16 spheroids. The SuperScript IV kit (Invitrogen, Waltham, MA, USA) was used for reverse transcription in accordance with the directions provided by the manufacturer. On a StepOnePlus instrument (Applied Biosystems/Thermo Fisher Scientific, Waltham, MA, USA), the Universal Taqman Master mix was employed to quantify the corresponding gene expressions. The amount of cDNA was standardized to the level of the housekeeping gene *36B4* (*Rplp0*) expression and is represented as a fold change from the control. Appendix A displays the primer and Taqman probe sequences that were employed.

### 2.5. Measurement of the Physical Properties, Size, and Stiffness of 3D HCSF Spheroids

The 3D sphenoid configurations were captured by using phase contrast (PC, Nikon ECLIPSE TS2; Tokyo, Japan), as previously mentioned, and the mean size of each 3D sphenoid, as determined by their largest cross-sectional area (CSA), was calculated by using Image-J software version 1.51n from the National Institutes of Health in Bethesda, MD, USA [33].

According to a recent study, the physical stiffness of the 3D spheroids was assessed by using a micro-squeezer (MicroSquisher, CellScale, Waterloo, ON, Canada) fitted with a 406 μm diameter cantilever microscale compression device [33]. Using a microscopic camera, a single organoid was put onto a 3 mm × 3 mm plate and compressed for 20 s to a 50% distortion. The cantilever was used to test the force necessary to produce a 50% strain under a variety of osmotic pressure settings, and the results are reported as force/displacement (N/m).

### 2.6. Statistical Analysis

GraphPad Prism 9 (GraphPad Software, San Diego, CA, USA) was utilized for all statistical analyses. A two-tailed Student’s *t*-test with a confidence level higher than 95% was employed to determine statistical significance for the comparison of two mean values. A grouped analysis with two-way analysis of variance (ANOVA) was carried out to examine group differences, and this was followed by a Tukey’s multiple comparison test. The arithmetic mean ± standard error of the mean (SEM) is how data are displayed.

## 3. Results

### 3.1. TGF-β2 Induced Effects on the Planar Proliferation of HCSFs

To study the TGF-β2 induced effects on the 2D cultured HCSFs, qPCR analysis for key ECM molecules such as *COL 1*: collagen 1; *COL 4*: collagen 4; *COL 6*: collagen 6; and *FN*: fibronectin was conducted. The other analyses included trans-endothelial electron resistance (TEER) measurements, FITC-dextran permeability, and scanning electronic microscope (SEM) investigations. As seen in Figure 1A, the treatment of 1 and 5 nM TGF-β2 resulted in an increase in the ECM deposits that covered the surface of the 100% confluent HCSF monolayers on day 6 of the experiment. In addition, the barrier function in the 2D HCSF monolayers, TEER values, and FITC-dextran permeability were significantly increased and decreased in the presence of TGF-β2, and this effect was concentration-dependent (Figure 1B,C). 

To support these results by SEM, TEER, and FITC-dextran permeability measurements, the mRNA expression of *COL1* and *COL4* and *FN* or *COL6* were substantially upregulated or downregulated, respectively, by TGF-β2 in a concentration-dependent manner (Figure 2). Furthermore, gene expressions of ECM regulating molecules such as the tissue inhibitor of metalloproteinase (*TIMP*) *1–4* and the matrix metalloproteinase (*MMP*) *2*, *3*, *9*, and *14* (Figure 3) and various ER stress-related factors were also significantly modulated. That is, as shown in Figure 3 and Figure 4, *TIMP 1* and *3*, *MMP 1*, and most ER stress-related factors or *MMP3* were significantly upregulated or downregulated, respectively, in a concentration-dependent manner. Therefore, these collective findings indicate that TGF-β2 significantly stimulated the expressions of most ECM molecules, their modulators, and ER stress-related factors. To support these findings, TGF-β2 also substantially increased both mitochondrial and glycolytic functions in the 2D HCSFs, suggesting that metabolic reprogramming was induced by the EMT induced by TGF-β2 (Figure 5).

### 3.2. TGF-β2 Induced Effects on the Spatial Proliferation of HCSFs

To study the TGF-β2-induced effects on the spatial proliferation of HCSFs, the 3D spheroid culture method was employed [32,33,36]. As shown in Figure 6, 3D HCSF spheroids were significantly downsized and stiffened by TGF-β2 in a concentration-dependent manner during the 6-day culture. However, in contrast to 2D HSCFs, the gene expressions of major ECM molecules (Figure 7), ECM modulators, TIMPs, and MMPs (Figure 8) were not significantly altered by TGF-β2 despite the finding that most of the ER stress-related factors were significantly or moderately upregulated (Figure 9). That is, upon TGF-β2 exposure, the expression of *COL1*, *COL6*, *TIMP2*, *MMP14* or *MMP2*, and *MMP3* were upregulated or downregulated, respectively. These collective results indicated that TGF-β2-induced effects toward the spatial proliferation of HCSFs were much lower than the planar proliferation of them, although those toward ER stress were comparable between both proliferations. 

## 4. Discussion

The cornea, the transparent outer layer of the human eye, consists of five distinct layers, epithelium, Bowman’s layer, stroma, Descemet’s membrane, and endothelium [37], and functions to protect the eyeball as well as regulate optical properties such as the transparency and refraction of the eye [38]. Among these five layers, the corneal stroma layer occupies approximately 90% of the thickness of the cornea and, thus, is the most critical structure for maintaining corneal transparency as well as protection. Therefore, once the corneal stroma is injured or scarred, serious visual deterioration could be evoked and may lead to blindness in the worst case [39]. To maintain this corneal transparency, both EMT and MET play pivotal roles [7] in which TGF-β2 signaling is known to be involved in their regulation [13]. That is, elevated TGF-β2 production in response to the IL-1a and IL-1β stimulation induces cell migration to corneal injury areas [40], and upregulation of ECM proteins, especially *COL4* expression in a wounded cornea [41]. In the current study, we also observed that TGF-β2 induced significant mRNA upregulations of *COL1*, *COL4*, *FN*, and *αSMA* in the 2D cultured HCSF cells (Figure 2). However, in contrast, such TGF-β2 induced upregulations of these ECM proteins were not observed in the 3D HCSF spheroids (Figure 7). As of this writing, we do not know why the mRNA of these ECM proteins was not upregulated upon exposure to TGF-β2 in the 3D HCSF spheroids. However, the relatively higher mRNA expression of *αSMA*, a marker of myofibroblast phenotype [14] of none-treated 3D HCSF spheroids and those that were not affected by TGF-β2 suggested that EMT may already be spontaneously evoked during 3D spheroid culture. This possibility is not surprising because such spontaneous biological activations during 3D spheroid cultures were also detected in the different cells. That is, during 3D spheroid culture, adipogenic differentiation of 3T3-L1 preadipocytes was spontaneously evoked [42], and spontaneous expressions of gap junction-related molecules in the H9c2 cardiomyocytes [43]. Therefore, this collective evidence indicated that such spontaneous biological activations may commonly occur beyond cell types and their origins, and in the case of HCSF, EMT may be easily induced within the spatial environment as compared with the planar environment. 

In vitro 3D models have been recognized as the preferable models for a better understanding of corneal wound healing, regeneration, and diseases because those models well replicate spatial tissue microenvironments on the cell behaviors, cell–cell interactions, and cell–matrix communications [44]. As compared to 2D cultures, previous studies indicated that the ECM molecules within a 3D culture more greatly influence cell proliferation, differentiation, and survival [45,46], and a 3D microenvironment closely resembles physiological access to nutrients, metabolites, oxygen, and signaling molecules [47]. Based upon these characteristics, in vitro 3D corneal models have been widely utilized for studying infection, injury, fibrosis, and regenerative mechanisms, as well as the screening of various drugs to evaluate their pharmacological effects and possible toxicities [48,49]. In fact, recently, we studied the roles of ROCK1 and 2 in the spatial architecture of human corneal stroma using a pan-ROCK inhibitor (pan-ROCK-i), ripasudil, and a ROCK2 inhibitor (ROCK2-i), KD025 and 3D spheroid models of HCSFs and found that ROCK1 and 2 differently influence the spatial architecture of 3D HCSF spheroids [31]. By using 2D and 3D cultures of HCSFs, we also examined the drug’s effects on the human corneal stroma, and the findings showed that the EP2 agonist omidenepag (OMD) modifies the physical stiffness of 3D HCSF spheroids in an osmotic pressure-dependent manner. Two-dimensional and three-dimensional cell cultures may be beneficial for assessing the drug-induced effects of OMD on human corneal stroma [32].

In conclusion, the findings presented in this study indicate that TGF-β2 induced EMT was quite different between planar and spatial directions within the corneal stroma, and, therefore, 3D HCSF spheroids may be a suitable in vitro model for replicating the 3D spatial architecture of the human corneal stroma for understanding the pathophysiological aspects of the corneal stroma. However, as the study limitation, differences in the biological roles of TGF-β2 and other TGF-β isoforms, TGF-β1 and TGF-β3 toward human corneal stroma have been insufficiently investigated so far. Therefore, to further elucidate the clinicopathological significance of these three TGF-β isoforms within human corneal stroma, additional investigations to compare TGF-β induced EMT among their three isoforms will be required using currently established in vitro 2D and 3D HCSF culture models as our next study project.

## Figures and Tables

**Figure 1 biomedicines-11-02513-f001:**
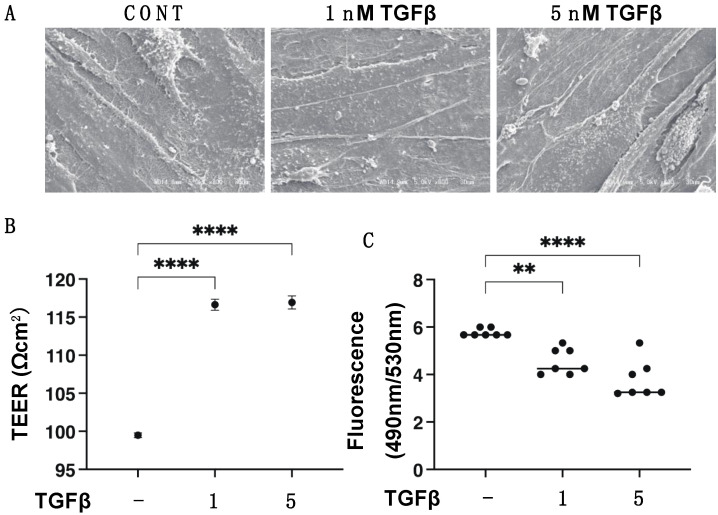
Effects of TGF-β2 on (**A**) ultrastructure by scanning electron microscopy (SEM), (**B**) trans-endothelial electrical resistance (TEER), and (**C**) the FITC-dextran permeability of 2D culture of human corneal stroma fibroblast (HCSF) monolayers. In the absence or presence of 1 nM or 5 nM TGF-β2, 2D cultured HCSF monolayers were prepared during a 6-day culture. Representative SEM images are shown in (**A**). Regarding their barrier functions, the TEER values (**B**) and FITC-dextran permeability (**C**) are plotted. All experiments were performed in duplicate using fresh preparations (n = 5, each). Data are presented as the arithmetic mean ± standard error of the mean (SEM). ** *p* < 0.01, **** *p* < 0.001 (ANOVA followed by a Tukey’s multiple comparison test).

**Figure 2 biomedicines-11-02513-f002:**
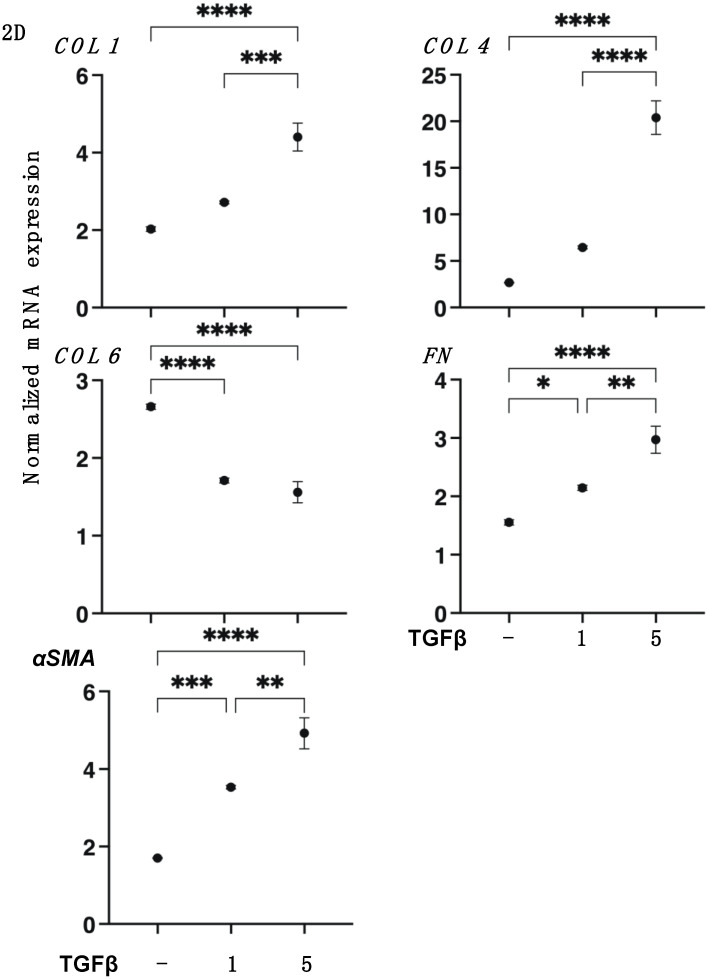
mRNA expression of major ECMs in 2D cultured human corneal stroma fibroblasts (HCSFs). During a 6-day culture, 2D cultured HCSF monolayers were created either in the absence or presence of 1 nM or 5 nM TGF-β2, and the ECM mRNA expression including *COL1*, *COL4*, *COL6*, *FN*, and αSMA was measured by qPCR analyses. Fresh preparations were used in duplicate for each experiment (n = 5, each). Data are presented as the arithmetic mean ± standard error of the mean (SEM). * *p* < 0.05, ** *p* < 0.01, *** *p* < 0.005, **** *p* < 0.001 (ANOVA followed by a Tukey’s multiple comparison test).

**Figure 3 biomedicines-11-02513-f003:**
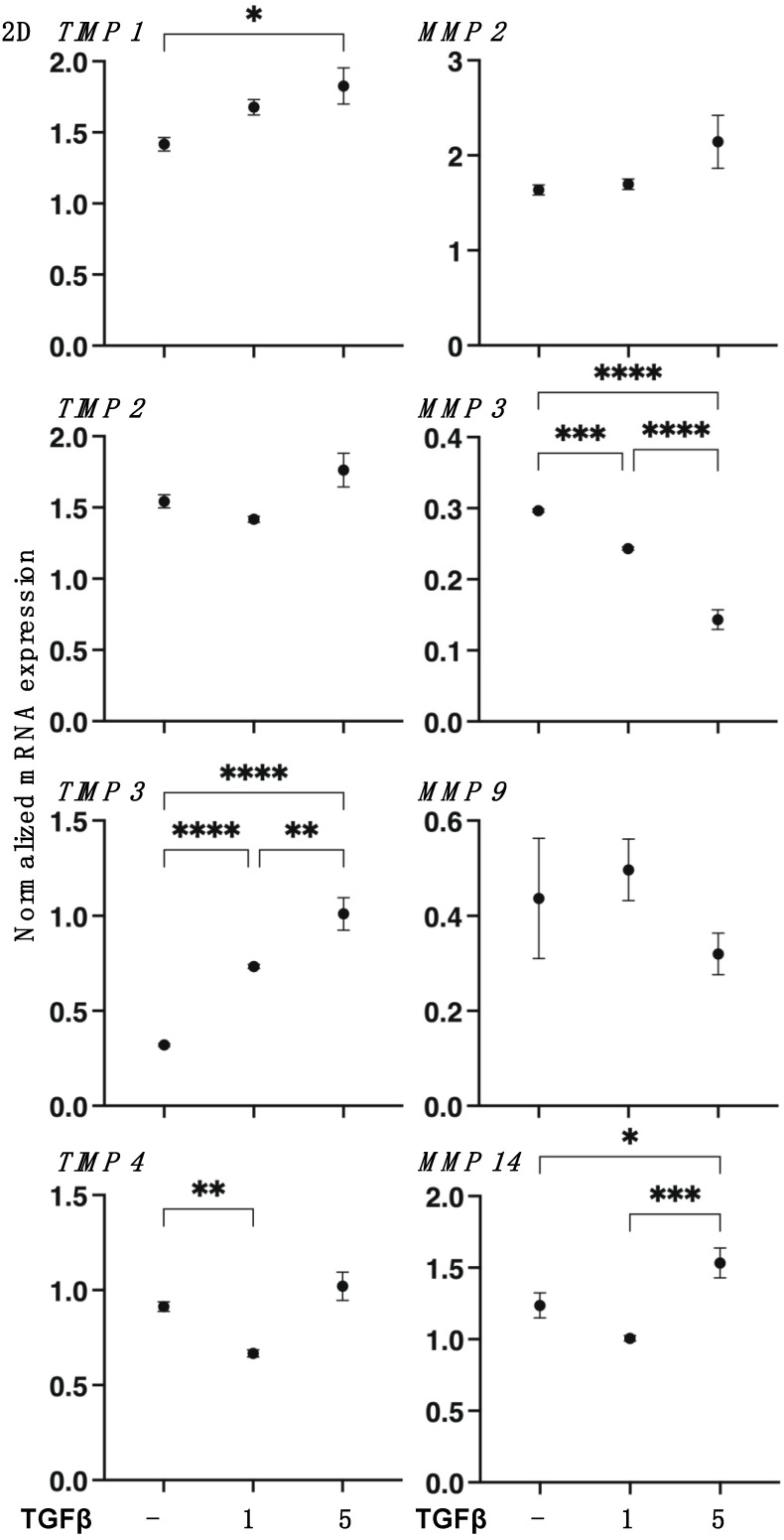
mRNA expression of *TIMPs* (*1–4*) and *MMPs* (*2*, *3*, *9*, *14*) in 2D cultured human corneal stroma fibroblasts (HCSFs). In the absence or presence of 1 nM or 5 nM TGF-β2, 2D cultured HCSF monolayers were prepared during a 6-day culture, and the mRNA expression of *TIMPs* (*1–4*) and *MMPs* (*2*, *3*, *9*, *14*) were measured by qPCR. All experiments were performed in duplicate using fresh preparations (n = 5, each). Data are presented as the arithmetic mean ± standard error of the mean (SEM). * *p* < 0.05, ** *p* < 0.01, *** *p* < 0.005, **** *p* < 0.001 (ANOVA followed by a Tukey’s multiple comparison test).

**Figure 4 biomedicines-11-02513-f004:**
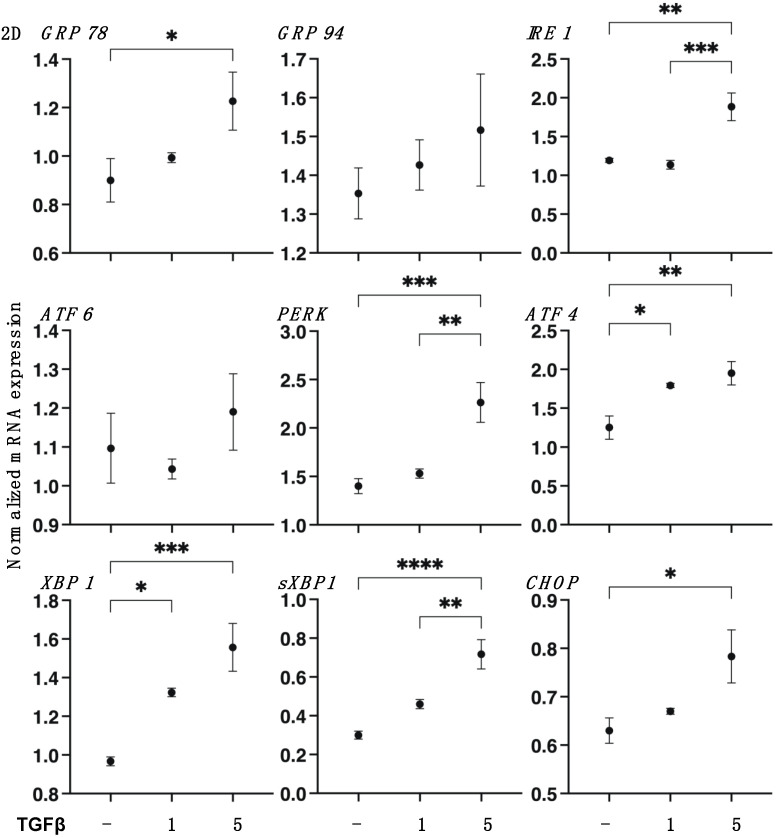
mRNA expression of several ER stress-related genes in 2D cultured human corneal stroma fibroblasts (HCSFs). In the absence or presence of 1 nM or 5 nM TGF-β2, 2D cultured HCSF monolayers were prepared during 6 days of culture and subjected to qPCR analysis to estimate the mRNA expression of several ER stress-related genes, including three principal regulators, i.e., the protein kinase RNA-like endoplasmic reticulum kinase (*PERK*), activating transcription factor 6 (*ATF6*) and the inositol-requiring enzyme 1 (*IRE1*), and their downstream factors including the glucose regulator protein (*GRP*)*78*, *GRP94*, the X-box binding protein-1 (*XBP1*), spliced XBP1 (*sXBP1*), and CCAAT/enhancer-binding protein homologous protein (*CHOP*). All experiments were performed in duplicate using fresh preparations (n = 5, in each setting). Data are shown as the arithmetic mean ± standard error of the mean (SEM). * *p* < 0.05, ** *p* < 0.01, *** *p* < 0.005, **** *p* < 0.001 (ANOVA followed by a Tukey’s multiple comparison test).

**Figure 5 biomedicines-11-02513-f005:**
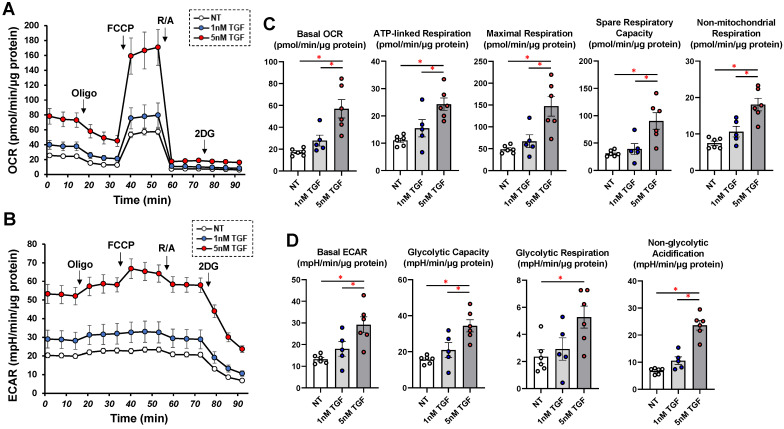
Effects of TGF-β2 on real-time cellular metabolic functions in 2D planar cultured human corneal stroma fibroblasts (HCSFs). During a 6-day culture, 2D cultured HCSF monolayers were created and put through real-time metabolic function measurement on a Seahorse XFe96 Bioanalyzer in the presence or absence of 1 nM or 5 nM TGF-β2. The rates of oxygen consumption rate (OCR, (**A**)) and extracellular acidification (ECAR, (**B**)) were assessed and were subsequently measured a second time after sequential supplementation with oligomycin (complex V inhibitor), FCCP (a protonphore), rotenone/antimyin A (complex I/III inhibitors), and 2DG (hexokinase inhibitor). (**C**,**D**) demonstrate important aspects of mitochondrial respiration and glycolytic flux, respectively. By deducting the baseline OCR from the OCR with rotenone/antimycin A, basal OCR was determined. ATP-linked respiration was then calculated by deducting OCR with oligomycin from OCR at the baseline. Maximal respiration was assessed by subtracting OCR with rotenone/antimycin A from OCR with FCCP. Spare respiratory capacity was assessed by deducting OCR at baseline from OCR with FCCP. Non-mitochondrial respiration was assessed by OCR with rotenone/antimycin A. Basal ECAR was assessed by deducting ECAR with 2DG from ECAR at baseline. Glycolytic capacity was assessed by deducting ECAR with 2DG from ECAR with oligomycin. The glycolytic reserve was calculated by deducting ECAR at baseline from ECAR with oligomycin. Non-glycolytic acidification was assessed by ECAR with 2DG. Non-mitochondrial respiration was calculated by OCR with rotenone/antimycin A. Basal ECAR was calculated by subtracting ECAR with 2DG from ECAR at baseline. Glycolytic capacity was calculated by subtracting ECAR with 2DG from ECAR with oligomycin. The glycolytic reserve was calculated by subtracting ECAR at baseline from ECAR with oligomycin. Non-glycolytic acidification was calculated by ECAR with 2DG. All experiments were performed using fresh preparations (n = 5–6). Fresh preparations were used in all studies (n = 5–6). Data are presented as the mean ± standard error of the mean (SEM). * *p* < 0.05 (ANOVA followed by a Tukey’s multiple comparison test).

**Figure 6 biomedicines-11-02513-f006:**
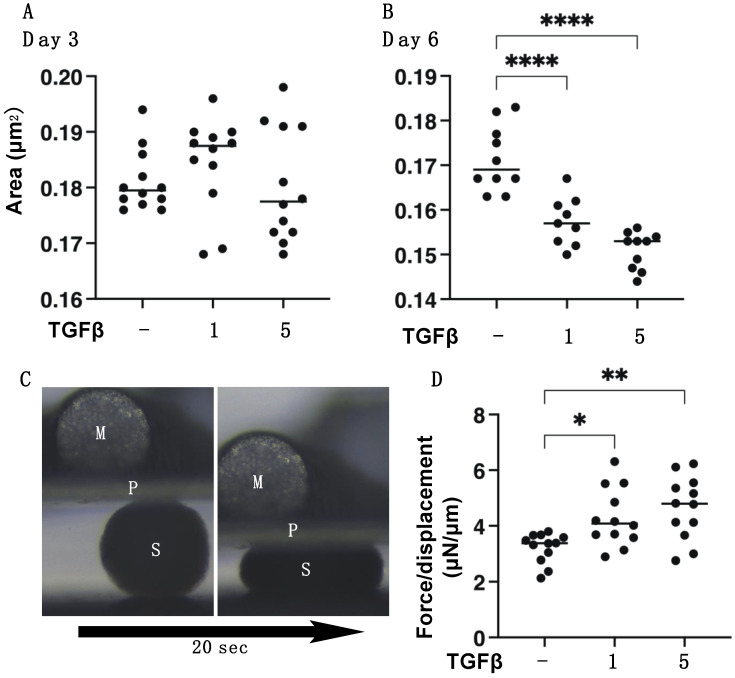
Effects of TGF-β2 on (**A**) the mean sizes of the 3D HCSF spheroids at day 3 and (**B**) day 6, and (**C**,**D**) physical solidity of the 3D HCSF spheroids on day 6. In the absence or presence of 1 nM or 5 nM TGF-β2, 3D HCSF spheroids were prepared during a 6-day culture, and their physical properties were determined. The mean sizes of the 3D HCSF spheroids at day 3 (**A**) and day 6 (**B**) are plotted (n = 10 each). (**C**) shows a physical stiffness analysis using a micro-squeezer; a single living 3D HCSFs spheroid at Day 6 was compressed until reaching its semi-diameter, and the force required (μN) was measured over a period of 20 s (S: 3D spheroid, P: compressing plate, M: micro-squeezer). The resulting force/displacement (μN/μm) values are plotted in (**D**) (n = 15 each). Data are presented as the arithmetic mean ± standard error of the mean (SEM). * *p* < 0.05, ** *p* < 0.01, **** *p* < 0.001 (ANOVA followed by a Tukey’s multiple comparison test).

**Figure 7 biomedicines-11-02513-f007:**
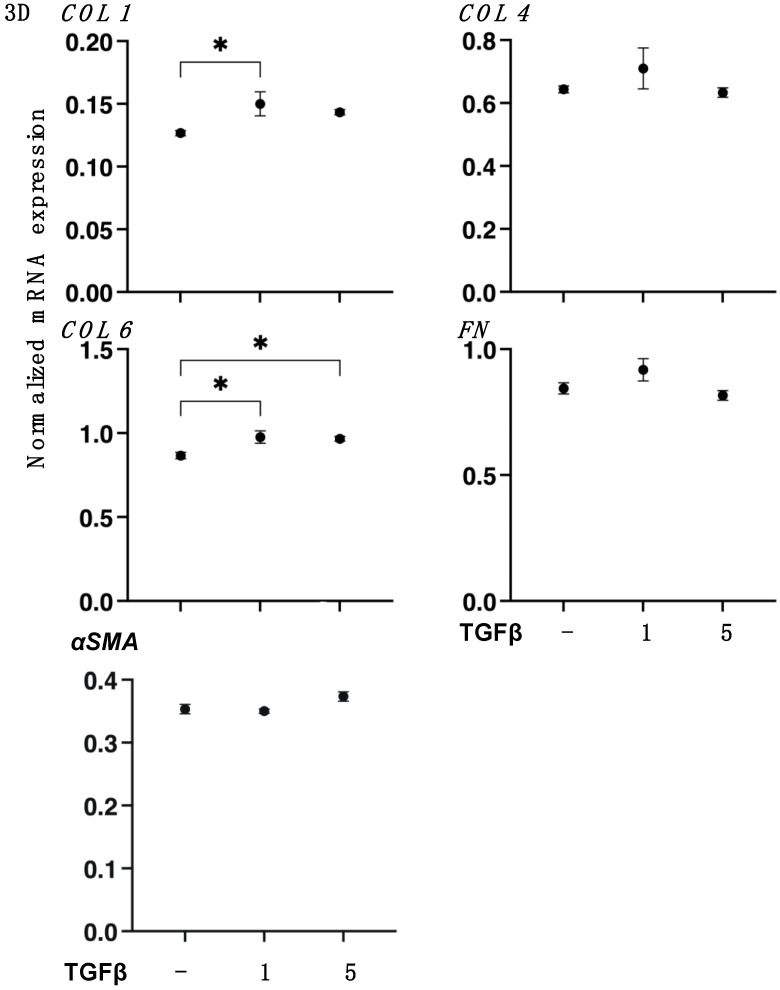
mRNA expression of major ECMs in 3D cultured human corneal stroma fibroblasts (HCSFs). In the absence or presence of 1 nM or 5 nM TGF-β2, 3D HCSF spheroids were prepared during a 6-day culture and assessed by qPCR analysis to evaluate the mRNA expression of ECM molecules including *COL1*, *COL4*, *COL6*, *FN*, and αSMA. Fresh preparations were used in duplicate for each experiment (n = 15, each). Data are presented as the arithmetic mean ± the standard error of the mean (SEM). * *p* < 0.05 (ANOVA followed by a Tukey’s multiple comparison test).

**Figure 8 biomedicines-11-02513-f008:**
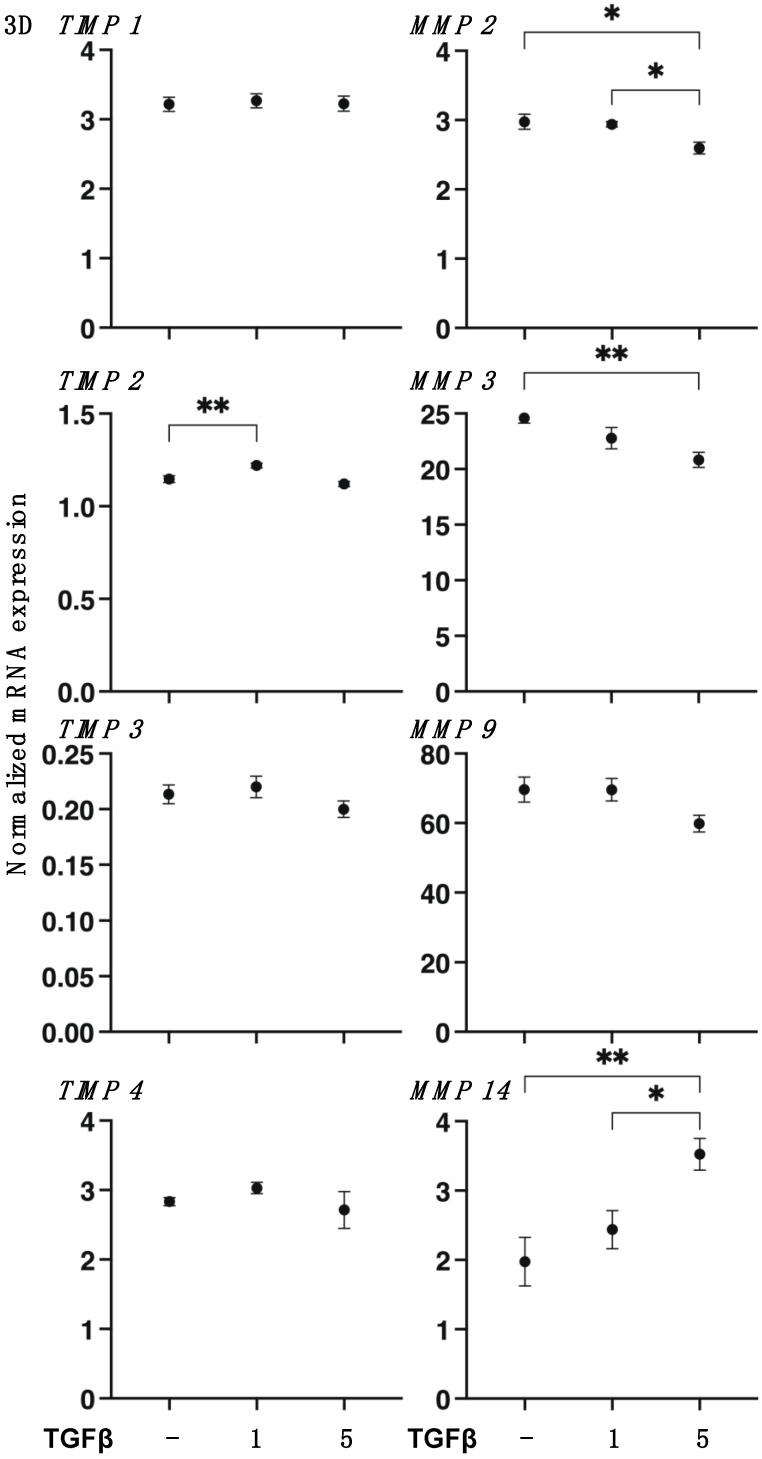
mRNA expression of TIMPs (1–4) and MMPs (2, 3, 9, 14) in 3D cultured human corneal stroma fibroblasts (HCSFs). In the absence or presence of 1 nM or 5 nM TGF-β2, 3D HCSF spheroids were prepared during a 6-day culture, and the mRNA expression of TIMPs (1–4) and MMPs (2, 3, 9, 14) was measured by a qPCR analysis. All experiments were performed in duplicate using fresh preparations (n = 15, each). Data are presented as the arithmetic mean ± standard error of the mean (SEM). * *p* < 0.05, ** *p* < 0.01 (ANOVA followed by a Tukey’s multiple comparison test).

**Figure 9 biomedicines-11-02513-f009:**
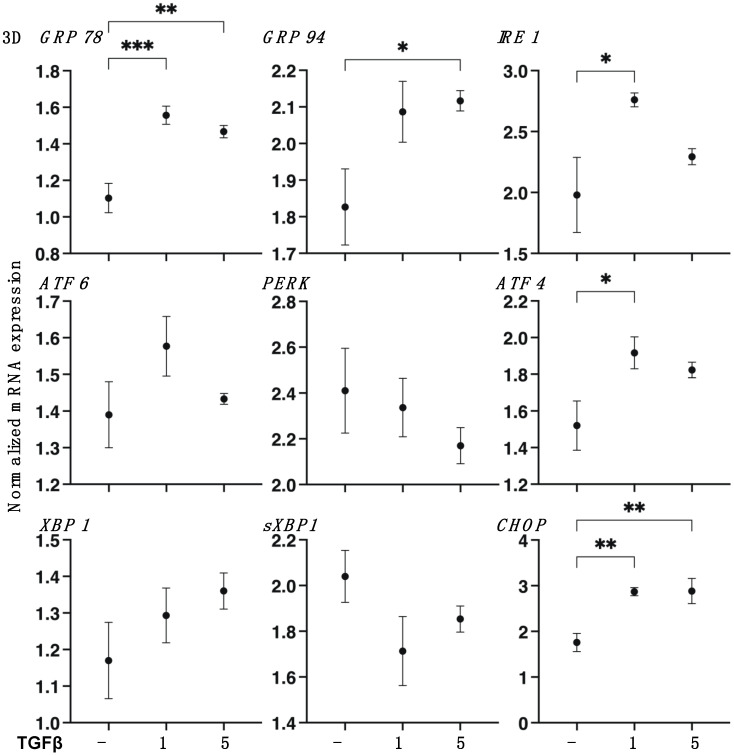
mRNA expression of several ER stress-related genes in 3D cultured human corneal stroma fibroblasts (HCSFs). In the absence or presence of 1 nM or 5 nM TGF-β2, 3D HCSF spheroids were prepared during 6 days of culture and subjected to qPCR analysis to estimate the mRNA expression of several ER stress-related genes including three master regulators, i.e., the protein kinase RNA-like endoplasmic reticulum kinase (*PERK*), activating transcription factor 6 (*ATF6*) and the inositol-requiring enzyme 1 (IRE1), and their downstream factors including the glucose regulator protein (*GRP*)*78*, *GRP94*, the X-box binding protein-1 (*XBP1*), spliced XBP1 (*sXBP1*), and CCAAT/enhancer-binding protein homologous protein (*CHOP*). All experiments were performed in duplicate using fresh preparations (n = 15, each condition). Data are presented as the arithmetic mean ± standard error of the mean (SEM). * *p* < 0.05, ** *p* < 0.01, *** *p* < 0.005 (ANOVA followed by a Tukey’s multiple comparison test).

## Data Availability

Data is contained within the article or Appendix A.

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
