# Peer review of "TGF-β2 Induces Epithelial–Mesenchymal Transitions in 2D Planer and 3D Spheroids of the Human Corneal Stroma Fibroblasts in Different Manners"

_biomedicines, 2023, doi:10.3390/biomedicines11092513_

Round 1

Reviewer 1 Report

Dear Authors,

Line 59:  "fibroblasts (19) and others" please add a reference to "and others". (authors could consider this article which evaluate the up-regulation of TGF beta in retinal disease: Coltrini D, Belleri M, Gambicorti E, Romano D, Morescalchi F, Krishna Chandran AM, Calza S, Semeraro F, Presta M. Gene expression analysis identifies two distinct molecular clusters of idiopatic epiretinal membranes. Biochim Biophys Acta Mol Basis Dis. 2020 Dec 1;1866(12):165938. doi: 10.1016/j.bbadis.2020.165938. Epub 2020 Aug 20. PMID: 32827649."

Line 336-337: Cornea has 5 layers (The corneal layers include epithelium, Bowman's layer, stroma, Descemet's membrane, and endothelium; Sridhar MS. Anatomy of cornea and ocular surface. Indian J Ophthalmol. 2018 Feb;66(2):190-194. doi: 10.4103/ijo.IJO_646_17. PMID: 29380756; PMCID: PMC5819093.) and not 3 as reported by Authors 

Author Response

Dear Editor,

Thank you very much for the constructive comments concerning our manuscript, " Physical properties and cellular metabolic characteristics of 3D spheroids are possible definitive indices for the biological nature of cancer-associated fibroblasts”. We carefully checked all of the Reviewer comments and prepared a revised version of our paper that takes these comments into account. The changes are listed below. Specific changes within the manuscript are highlighted.

Reviewer 1

Dear Authors,

  1. Line 59: "fibroblasts (19) and others" please add a reference to "and others". (authors could consider this article which evaluate the up-regulation of TGF beta in retinal disease: Coltrini D, Belleri M, Gambicorti E, Romano D, Morescalchi F, Krishna Chandran AM, Calza S, Semeraro F, Presta M. Gene expression analysis identifies two distinct molecular clusters of idiopatic epiretinal membranes. Biochim Biophys Acta Mol Basis Dis. 2020 Dec 1;1866(12):165938. doi: 10.1016/j.bbadis.2020.165938. Epub 2020 Aug 20. PMID: 32827649."

Answer; Thank you so much for this excellent comment. As suggested this reference is included.

  1. Line 336-337: Cornea has 5 layers (The corneal layers include epithelium, Bowman's layer, stroma, Descemet's membrane, and endothelium; Sridhar MS. Anatomy of cornea and ocular surface. Indian J Ophthalmol. 2018 Feb;66(2):190-194. doi: 10.4103/ijo.IJO_646_17. PMID: 29380756; PMCID: PMC5819093.) and not 3 as reported by Authors

Answer; Thank you so much for this excellent comment. As suggested, corneal layers were corrected with this reference.

Reviewer 2 Report

authors wrote an interesting article

I suggest few improvements:

1) introduction: 

Upon arrival at the injured corneal stroma, the fibroblasts differ-36entiate into myofibroblasts that elaborate extracellular matrix (ECM) molecules and gen- 37 erate contractile forces that are engaged in the closure of the corneal wound (4).

Here please add some clinical examples, like recovery from corneal ulcers

Please have a look and cite the following paper in introduction, they may help:

  • PMID: 30681513

  • PMID: 29800578

Please improve the limitations of the study

English is fine

Author Response

Dear Editor,

Thank you very much for the constructive comments concerning our manuscript, " Physical properties and cellular metabolic characteristics of 3D spheroids are possible definitive indices for the biological nature of cancer-associated fibroblasts”. We carefully checked all of the Reviewer comments and prepared a revised version of our paper that takes these comments into account. The changes are listed below. Specific changes within the manuscript are highlighted.

Reviewer 2

authors wrote an interesting article

I suggest few improvements:

  1. introduction: Upon arrival at the injured corneal stroma, the fibroblasts differ-36entiate into myofibroblasts that elaborate extracellular matrix (ECM) molecules and gen- 37 erate contractile forces that are engaged in the closure of the corneal wound (4).Here please add some clinical examples, like recovery from corneal ulcers. Please have a look and cite the following paper in introduction, they may help: PMID: 30681513     PMID: 29800578

Answer; Thank you for this excellent comment. As suggested, 1st paragraph of the introduction was changed with suggested references; “In the space of injured corneal stroma, it is well known that the keratocytes change to spindle-shaped fibroblasts and acquire a migratory phenotype with the increased actin expression. This increment in actin expression enables fibroblasts to proliferate and migrate towards the wound, repopulating the region that had been depleted of keratocytes through apoptosis (1-3). Upon arrival to the injured corneal stroma, the fibroblasts differentiate into myofibroblasts that elaborate extracellular matrix (ECM) and generate contractile forces engaged in corneal wound closure (4). Clinically, such corneal stromal recovery from corneal damages was indeed observed in some cases surgically treated using simple limbal epithelial transplantation (5). In addition, within the biobanked eyes, the molecular or biomechanical properties of human donor corneal stroma can be restored by rehydration after a dehydrated condition for up to 2 weeks (6).”.

  1. Please improve the limitations of the study

Answer; Thank you for this excellent comment. As suggested, last paragraph of discussion was rewritten including study limitation; “In conclusion, the findings presented in this study indicate that TGF-b-2 induced EMT was quite different between planar and spatial directions within the corneal stroma, and therefore, 3D HCSFs spheroids may be a suitable in vitro model for replicating the 3D spatial architecture of the human corneal stroma for understanding the pathophysiological aspects of the corneal stroma. However, as the study limitation, differences of the biological roles of TGF-b-2 and other TGF-b isoforms, TGF-b-1 and TGF-b-3 toward human corneal stroma have insufficiently investigated so far. Therefore, to elucidate further the clinicopathological significance of these three TGF-b isoforms within human corneal stroma, additional investigations to compare TGF-b induced EMT among their three isoforms will be required using currently established in vitro 2D and 3D HCSFs culture models as our next study project.”.

Round 2

Reviewer 1 Report

Dear Authors,

Thanks for the amendments.

Reviewer 2 Report

accepted